# LIGHT: LLM-GUIDED GRAPH EXPERT ROUTING FOR SEMI-SUPERVISED DOMAIN GENERALIZATION

## ABSTRACT

Although graph neural networks (GNNs) have shown remarkable performance in graph machine learning, their effectiveness in practice often suffers from realistic challenges including distribution shifts and label scarcity. Towards this end, this paper studies the problem of semi-supervised domain generalization, which aims to improve the performance of GNNs on unseen graphs using both labeled and unlabeled data. We propose a novel approach named LLM-Guided Graph Expert Routing (`LIGHT`) for semi-supervised domain generalization. The core idea of `LIGHT` is to distill the knowledge from LLM-as-a-judge to determine context-aware routing weights for a multi-hop graph mixture-of-experts framework. In particular, our `LIGHT` employs diverse graph experts that explore neighborhood information at varying depths. More importantly, we leverage LLMs to provide judgments of the most reliable graph experts are for crucial nodes, which provide context-aware routing guidance with high generalizability for knowledge distillation. To further address label scarcity, we introduce an expert-aware dynamic pseudo-labeling strategies that selects reliable nodes for additional training. Extensive experiments on various benchmark datasets validate the effectiveness of the proposed `LIGHT` in comparison with competing approaches. Our source code can be found at https://anonymous.4open.science/r/LIGHT-A817.

## 1 INTRODUCTION

Graphs serve as a fundamental data structure for capturing intricate relational and structural dependencies. With their proven effectiveness and computational efficiency in graph-based machine learning, Graph Neural Networks (GNNs) have become a *de facto* apporach across a variety of tasks. One fundamental task is node classification, which endeavors to predict the label of each node within a graph based on its features and relational context, has been widely applied in various domains, i.e., social network analysis Bhagat et al. (2011), biochemistry Yue et al. (2020) and recommender systems Wu et al. (2022b). However, GNNs inherently rely on the assumption that training and testing data are independently and identically distributed (i.i.d.) Pan & Yang (2010), making them susceptible to the out-of-distribution (OOD) problem that induces significant performance drops under *distribution shifts*.

To prevent the generalization failure of GNNs, recent efforts have focused on the invariance principle from causality and invariant learning has emerged as a promising strategy for generalizable graph learning. Actually, GNNs' OOD generalization (domain generalization) is impaired by spurious associations between non-essential graph patterns and target labels Chen et al. (2024b); Ju et al. (2024b). The rationale of invariant graph learning is to identify the invariant subgraph that consistently contributes to the prediction under distribution shift Li et al. (2022b); Wu et al. (2022a); Liu et al. (2023a). Despite the effectiveness, these methods often assume fully labeled source domains, whereas in real-world scenarios, the source domain typically contains only a small number of labeled samples alongside a large amount of unlabeled data Qiao et al. (2023b); Dan et al. (2024a;b).

In this work, we focus on a more realistic problem of semi-supervised domain generalization on graphs. Nonetheless, straightly adapting the above methods remains non-trivial, primarily arising from two key challenges: **First**, *The availability of diverse training environments under label scarcity setting.* Effective disentanglement of invariant components from spurious correlations re-

quires exposure to a variety of training environments Xia et al. (2023); Li et al. (2023); Wang et al. (2024b). However, in real-world settings, unseen domains often involve a broad spectrum of latent factors, whose quantity and nature are difficult to specify and replicate. This issue is further exacerbated when labeled data are scarce, making it difficult to reliably simulate the necessary diversity for invariant learning. **Second**, *The heterogeneity of graph data could degrade generalization ability.* Contrary to regular Euclidean data, graphs inherently possess diverse and intricate structures, with each node connected to different neighbors, resulting in highly divergent neighborhood structures and node degrees. The message-passing mechanism in GNNs tends to emphasize signals from nodes or subgraphs with high degrees and strong modularity Liu et al. (2023b); Sun et al. (2024). This often induces a simplicity bias, where the model focuses on prominent yet potentially spurious patterns, at the expense of capturing the true invariant substructures critical for robust prediction.

Fortunately, the remarkable generalization capabilities of Large Language Models (LLMs) provide new avenues for addressing these challenges. By harnessing their extensive pre-trained knowledge and semantic reasoning abilities, LLMs can offer valuable guidance to complement the learning process of GNNs. On the one hand, some prior works have utilized LLMs as enhancers to enrich node or graph representations Chien et al. (2022); Jin et al. (2023); He et al. (2024); Liu et al. (2024); Li et al. (2024a). On the other hand, some approaches have attempted to directly employ LLMs as predictors Zhao et al. (2023); Fatemi et al. (2024); Tang et al. (2024); Perozzi et al. (2024); Chen et al. (2024a); Wang et al. (2024a). However, LLMs are fundamentally designed to generate discrete textual outputs in a decoder-only manner, which limits their ability to accurately represent complex graph semantics Wei et al. (2022); Jiang & Luo (2025). Meanwhile, directly generating predictions from LLMs often leads to instability and unrelated outputs required for graph-based tasks.

Towards the end, in this paper, we propose a novel approach named **L**LM-gu**I**ded **G**rap**H** Mixture-of-expert Rou**T**ing (`LIGHT`) for semi-supervised domain generalization on graphs, which aims to utilize LLMs as a context-aware routing to select appropriate invariant subgraph experts under the label scarcity setting. Specifically, given the input graph, we first transform the context of the node from different hops (i.e., 0-hop, 1-hop and 2-hop) into prompts, which are fed into pre-trained LLMs. Then, instead of directly predicting node labels, our `LIGHT` employs a graph mixture-of-expert (MoE) framework Wang et al. (2023), where each expert is dedicated to handling distinct data distributions, thereby facilitating better alignment with invariant correlations. Based on this, we generate predictions from each graph expert and LLMs are employed to dynamically select the most contextually aligned output. Finally, pseudo-labels are assigned to the remaining unlabeled nodes, and a flexible confidence threshold informed by expert outputs is utilized to identify high-confidence pseudo-labeled nodes. Therefore, benefiting from the zero-shot capabilities of LLMs, the framework could generalize effectively to unseen domains under the semi-supervised scenario.

The contribution of our `LIGHT` can be summarized as follows:

❶ *New Perspective.* We study an underexplored yet practical problem of semi-supervised domain generalization on graphs and pioneer a path of leveraging zero-shot capabilities of LLMs to address this problem.

❷ *Novel Methodology.* We propose a novel LLM-Guided MoE framework `LIGHT`, which transforms the multi-hop ego-subgraphs of each node into LLM prompts to guide the selection of GNN expert as the invariant predictor, and introduces a flexible expert-enhanced pseudo-labeling strategy to supervise the training of unlabeled nodes.

❸ *Extensive Experiments.* We conduct extensive experiments on multiple public datasets to evaluate the effectiveness of the proposed framework. The results consistently demonstrate the superiority of `LIGHT` in semi-supervised graph domain generalization.

## 2    RELATED WORK

**Graph Domain Generalization.** Graph Domain Generalization (GDG) has emerged as a critical research area to address the prevalent issue of performance degradation when GNNs encounter OOD data Li et al. (2022a); Mai et al. (2025). Early efforts focused on self-supervised pretraining (e.g. Pretraining-GNN), disentanglement (e.g. DisenGCN Ma et al. (2019a)), and adversarial training (e.g. GraphAT Zhang et al. (2025). Recently, casual inference with representative methods including CIGA Chen et al. (2022) and StableGNN Polina Andreeva & Bochenina (2022). Data augmentation techniques have matured and label semantic preservation (e.g. LiSA Yu et al. (2023)). Complex multicomponent frameworks, such as MLDGG Tian et al. (2025) which integrates meta-learning

and causal reasoning, and TRACI Zhao et al. (2025a) which employs topological adversarial learning and prototypical mixup, signify a shift towards holistic solutions. However, these methods often assume fully labeled source domains, which limits their applicability in real-world scenarios where label annotations are scarce or expensive. In our paper, we consider the graph domain generalization task under the semi-supervised scenario.

**Semi-supervised Learning on Graphs.** Semi-Supervised Learning on Graphs (GSSL) has emerged as a crucial paradigm for learning from graph-structured data where labeled instances are scarce but unlabeled data and structural information are abundant Zhao et al. (2025b). Traditional GSSL includes Label Propagation Algorithms Kipf & Welling (2017); Ju et al. (2024a) and graph regularization techniques that enforce smoothness in predictions, while current emphasizes enhancing robustness and scalability through regularization Wu et al. (2025); Parveen et al. (2024), pseudo-labeling, and sophisticated Graph Contrastive Learning coupled with advanced data augmentation strategies Du et al. (2025); Wang et al. (2025); Ma et al. (2019b). Build on various kind on GNNs architecture, advanced SSL paradigms emerged, self-training strategies like A3-GCN Abdolali et al. (2025) and graph contrastive learning method like DGI Zhu et al. (2020), GRACE Hu et al. (2024) and AFECL Li et al. (2024b) for unsupervised representation learning. However, how to effectively perform semi-supervised learning under distribution shift remains underexplored.

# 3 PRELIMINARIES

**Notations.** Let $\mathcal{G} = \{\mathcal{V}, \mathcal{E}\}$ denote a graph, where $\mathcal{V}$ is the node set with $N$ nodes and $\mathcal{E}$ is the edge set. The adjacency matrix is denoted as $\boldsymbol{A} \in \{0,1\}^{N \times N}$, where $\boldsymbol{A}_{uv} = 1$ if there is an edge $(u,v) \in \mathcal{E}$, otherwise $\boldsymbol{A}_{uv} = 0$. Besides, the node feature matrix can be denoted as $\boldsymbol{X} \in \mathbb{R}^{N \times d}$, where $d$ is the feature dimension. From the data-generating perspective, the input graph can be decomposed into a set of overlapping ego-graphs Wu et al. (2022a; 2024b) and the $k$-hop ego-graph of node $v$ can be defined as $\mathcal{G}_v^{(k)} = \{\boldsymbol{X}_v^{(k)}, \boldsymbol{A}_v^{(k)}\}$, where feature matrix $\boldsymbol{X}_v^{(k)}$ and adjacency matrix $\boldsymbol{A}_v^{(k)}$ is induced by the node $v$ as well as its $k$-hop neighborhood respectively.

**Semi-supervised Domain Generalization on Graphs.** Given the graph $\mathcal{G} = \{\mathcal{V}, \mathcal{E}\}$, we denote $\mathcal{V}^s \subset \mathcal{V}$ as the labeled node set, where each node $v \in \mathcal{V}^s$ is assigned a label $\boldsymbol{Y}_v \in \{0,1\}^C$, with $C$ being the number of classes, and $\boldsymbol{Y}_{v,c} = 1$ if node $v$ belong to $c$-th class, otherwise $\boldsymbol{Y}_{v,c} = 0$. The rest $\mathcal{V}^u = \mathcal{V} \backslash \mathcal{V}^s$ is an unlabeled node set. The distribution shifts on the graph indicate that the distribution of ego-subgraphs and their labels differs between the training and testing data, namely $p_{tr}(\mathcal{G}_v, y_v) \neq p_{te}(\mathcal{G}_v, y_v)$. The objective is to learn an optimal predictor $f^*(\cdot)$ given the source graph $\mathcal{G}$ with limited labels such that it performs well on the unseen target graph.

$$f^* = \arg \min_f \mathbb{E}_{p_{tr}}[\mathcal{L}(f(\mathcal{G}_v), \boldsymbol{Y}_v)], \tag{1}$$

where $\mathcal{L}(\cdot)$ is the loss function. The learned model is then utilized to infer the node label for the target graph. Note that here the source and target graphs share the same label space.

# 4 METHODOLOGY

## 4.1 FRAMEWORK OVERVIEW

We introduce an LLM-guided MoE framework for semi-supervised domain generalization on graphs. The high-level idea of our `LIGHT` is to leverage the zero-shot capability of LLMs to dynamically select appropriate GNN experts based on multi-hop ego-subgraph prompts, thereby enhancing the generalization to unseen domains. In particular, we first decouple the node context into different hops and feed them into an MoE framework. Based on this, we extract the context of each node from different hops into prompts and integrate them into LLMs to adaptively route inputs to the most relevant experts with high generalizability for knowledge distillation. Finally, we utilize an expert-enhanced pseudo-labeling to alleviate the label scarcity. The overview of our `LIGHT` is illustrated in Figure 1. More details are introduced as follows.

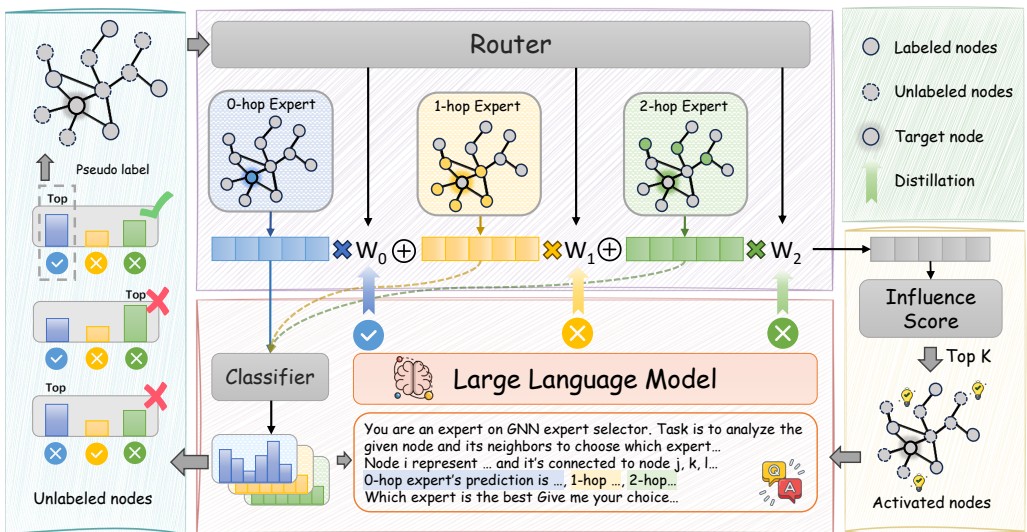

Figure 1: The overview framework of our proposed LIGHT. We decouple the node context into different hops and feed them into a mixture-of-expert framework. Then, we transform the context into prompts and input them into an LLM. The LLMs serve as a teacher router and we distill the knowledge from LLM-as-judge. Finally, we introduce expert-aware dynamic pseudo-labeling to select reliable nodes for additional training.

## 4.2 MULTI-HOP GRAPH MIXTURE-OF-EXPERT FOR DIVERSE NEIGHBORHOOD EXPLORATION

Traditional GNNs often suffer from oversmoothing or overfitting when aggregating multi-hop neighbors, particularly under distribution shifts. These issues can be attributed to the indiscriminate aggregation of neighborhood information across varying hops. This practice can lead to the generation of spurious correlations or redundant patterns both of which are sensitive to domain-specific biases. Theorem 4.1 demonstrates that the mixture-of-experts (MoE) model achieves a lower prediction error compared to the individual model.

**Theorem 4.1.** *Suppose the true label for node $v$ is 1. If only a single expert is available (i.e., $K = 1$) and this expert is not aligned with the domain-specific distribution of node $v$, then the prediction error satisfies the lower bound:*

$$\mathbb{P}(\hat{\boldsymbol{Y}}_v \neq 1) = \mathbb{P}(Q_{v,1} < 0) > 0.$$

*In contrast, if $K$ experts are available, the prediction error is upper bounded as:*

$$\mathbb{P}(\hat{\boldsymbol{Y}}_v \neq 1) \leq \frac{C_0}{K},$$

*where $C_0$ is a constant that depends on the margin $\nu$ and the sub-Gaussian parameter $\sigma^2$.*

The proof of this bound is provided in Appendix B. Notably, different hops capture different levels of contextual semantics: lower hops (e.g., 1-hop) typically reflect local homophilic patterns, while higher hops encode global and heterophilic dependencies. Therefore, we decouple information aggregation by hop and generate a set of candidate predictions by designing a graph MoE framework Cai et al. (2024); Wang et al. (2023), where hop-specific experts generate candidate predictions that are adaptively integrated based on node context for robust performance under distribution shift.

For the $k$-hop ego-subgraph $\mathcal{G}_v^{(k)}$, we calculate positive pointwise mutual information (PPMI) Zhuang & Ma (2018) between nodes to fully explore the structural information. Specifically, we employ a random walk to sample a set of paths on $\boldsymbol{A}$, and the frequency matrix $\boldsymbol{F}$ between

nodes can be calculated as the co-occurrence on the paths. Then, the PPMI between nodes can be:

$$\tilde{\boldsymbol{P}}_{ij} = \frac{\boldsymbol{F}_{ij}}{\sum_{i,j} \boldsymbol{F}_{ij}}, \tilde{\boldsymbol{P}}_i = \frac{\sum_j \boldsymbol{F}_{ij}}{\sum_{i,j} \boldsymbol{F}_{ij}}, \tilde{\boldsymbol{P}}_j = \frac{\sum_i \boldsymbol{F}_{ij}}{\sum_{i,j} \boldsymbol{F}_{ij}},$$

$$\boldsymbol{P}_{ij} = \max\{\log(\frac{\tilde{\boldsymbol{P}}_{ij}}{\tilde{\boldsymbol{P}}_i \times \tilde{\boldsymbol{P}}_j}), 0\}. \tag{2}$$

where $\tilde{\boldsymbol{P}}_{ij}$ denotes the probability that node $v_i$ within the contextual window of node $v_j$ and $\boldsymbol{P}_{ij}$ is the PPMI between them. Therefore, a larger $\boldsymbol{P}_{ij}$ indicates a higher co-occurrence frequency between nodes than if they were independent. We treat $\boldsymbol{P}_v^{(k)}$ as the new adjacency matrix for the $k$-hop ego-subgraph $\mathcal{G}_v^{(k)}$ and the representation of node $v$ can be extracted as:

$$\boldsymbol{Z}_v^{(k)} = \text{GNN}_k(\boldsymbol{X}_v^{(k)}, \boldsymbol{P}_v^{(k)}), \tag{3}$$

where $\text{GNN}_k(\cdot)$ is the GNN with $L$ layers. Note that in our work, we employ $K$ experts sharing the same architecture but parameterized independently. We utilize a MoE routing mechanism to assign expert weights, followed by a classifier that produces the final prediction:

$$\hat{\boldsymbol{Y}}_v = f\Big(\sum_{k=1}^{K} \boldsymbol{\omega}_v^{(k)} \boldsymbol{Z}_v^{(k)}\Big), \tag{4}$$

where $f(\cdot)$ is a learnable classifier and $\boldsymbol{\omega}_v$ denote the weight of the different expert.

### 4.3 Neighborhood-aware Prompt Engineering

Since the distribution shifts on the graph entail a broad range of structural variations, directly learning a routing function for $\boldsymbol{\omega}_v$ as in traditional MoE frameworks becomes unreliable. We leverage the zero-shot capability of LLMs as a judge to enable more informed and adaptive expert selection. In particular, for each node $v$, we first extract $K$ ego-subgraphs corresponding to different hops, effectively capturing multi-hop structural contexts. Then, we take each ego-graph $\mathcal{G}_v$ as input of the expert and provide the context to describe the expert ($g : \mathcal{G}_v \to \boldsymbol{W}$) and the goal of tasks ($q : \boldsymbol{W} \to \boldsymbol{W}$), where $\boldsymbol{W}$ is the sequence of discrete tokens for the LLMs.

**Expert Context Provision.** Given a node $v$, the ego-subgraph for each expert and its context can be presented as: *You are an expert on GNN experts selector, given GNN experts:* $\langle$expert$\rangle$...\n, with *node $v$ and its neighborhood* {graph} *represent node connection relationships...\n*, where $\langle$expert$\rangle$ is the placeholder for expert inputs and {graph} is the ego-subgraph information.

Note that the $k$-th expert input is the description w.r.t. $k$-hop neighborhood, i.e., the 1-hop expert can be *1-hop: direct connected neighbors providing classification signals*. Ego-subgraph information can be described in an incident manner, i.e., *Node $v$ represents..., and it is connected to node $u$...\n*.

**Task Description.** To guide the expert selection process, we design a task-specific prompt where the model is asked to choose the most appropriate expert for node classification based on the expert context. Specifically, the prompt includes not only the task description but also a set of alternative expert choices, i.e., *Which expert does node $v$ classification used? Please give out your choice on the following experts:* {ans}, where {ans} represent the set of experts, namely $\{1, 2 \ldots, K\}$.

Based on this, the context information of nodes can be formulated as prompts $(g(\mathcal{G}_v), q(\boldsymbol{Q}), \mathcal{R}_v)$, where $\boldsymbol{Q}$ denote the question text for the task. Note that we also incorporate the individual predictions $\mathcal{R}_v$ from each expert as the context (see Section 4.2).

### 4.4 LLM-guided Context-aware Routing with Knowledge Distillation

Instead of relying on LLMs to directly generate predictions, which often results in unreliable outputs or incorrect formats, we propose to leverage LLMs as a judge instead of a predictor and employ knowledge distillation to transfer the LLM-guided routing behavior to a lightweight student model.

**LLM Judge for Context-aware Routing.** We generate the predictions from different experts and incorporate them into the LLM judge for contextual-aware routing, which can be formulated as:

$$\mathcal{R}_v = \{\boldsymbol{R}_v^{(k)}\}_{k=1}^{K}, \ \boldsymbol{R}_v^{(k)} = f\Big(\sum_{k'=1}^{K} \boldsymbol{e}_v^{(k)}(k') \boldsymbol{Z}_v^{(k)}\Big), \tag{5}$$

where $\boldsymbol{e}_v^{(k)}$ is initialized as a one-hot vector with the $k$-th entry set to 1. Then, the resulting predictions are subsequently incorporated into the LLMs, which selects the most plausible expert $k^*$:

$$k^* = \text{LLM}(g(\mathcal{G}_v), q(\boldsymbol{Q}), \mathcal{R}_v). \tag{6}$$

We adopt a label smoothing strategy Müller et al. (2019) within our MoE framework by assigning lower weights to non-selected experts. Accordingly, the weight vector $\boldsymbol{e}_v^{(k)}$ can be updated as:

$$\boldsymbol{e}_v^{(k)} = \begin{cases} \alpha, & \text{if } k = k^*, \\ \frac{1-\alpha}{K-1}, & \text{if } k \neq k^*, \end{cases} \tag{7}$$

where $\alpha \in (0, 1)$ is the smoothing coefficient.

**Knowledge Distillation with Influence Maximization.** Note that performing LLM inference for each node is computationally inefficient, we distill a lightweight routing function from the LLM to enable efficient expert selection. Inspired by the influence maximization (IM) problem in the social network Kempe et al. (2003), we aim to select a set of anchor nodes $\mathcal{S}$ that maximizes the utility of LLMs in a cost-effective manner, and the formation is:

$$\max_{\mathcal{S}} |\sigma(\mathcal{S})| \text{ s.t. } \mathcal{S} \subseteq \mathcal{V}, |\mathcal{S}| = B, \tag{8}$$

where $\sigma(\mathcal{S})$ denotes the set of nodes that can be activated by $\mathcal{S}$, defined as:

$$\sigma(\mathcal{S}) = \bigcup_{v \in \mathcal{V}, I_v(\mathcal{S}) > \theta} \{v\}, \ I_v(\mathcal{S}) = \max_{u \in \mathcal{S}} I_v(u), \tag{9}$$

where $I_v(\mathcal{S})$ denotes the maximum influence from the anchor nodes, $\theta$ is the activation threshold. In practice, we introduce $\boldsymbol{P}_{u,v}$ and the influence quality $I_v(u)$ from anchor node $u$ is measured as:

$$I_v(u) = \boldsymbol{P}_{u,v} \sum_{k=1}^{K} E_u^{(k)}, \tag{10}$$

where we use entropy $E_u^{(k)} = -\sum_{c=1}^{C} \boldsymbol{R}_{u,c}^{(k)} \log \boldsymbol{R}_{u,c}^{(k)}$ as the influence score. We employ a greedy node selection to obtain $\mathcal{S}$ and the details can be seen in Appendix 1. Based on $\mathcal{S}$, we distill knowledge from the LLMs to obtain a lightweight routing function, and the teaching loss can be:

$$\mathcal{L}_{dl} = -\sum_{u \in \mathcal{S}} \text{CE}(\boldsymbol{e}_u^{(k^*)}, h(\boldsymbol{Z}_u)), \ \boldsymbol{Z}_u = \boldsymbol{Z}_u^{(1)} \| \ldots \| \boldsymbol{Z}_u^{(K)}, \tag{11}$$

where $h(\cdot)$ denotes the routing function and $\text{CE}(\cdot)$ is the cross-entropy loss. Note that to effectively address domain shifts, we perform knowledge re-distillation from the LLMs for each target domain prior to inference, and the MoE framework in Equation 3 can be updated into:

$$\hat{\boldsymbol{Y}}_v = f\Big(\sum_{k=1}^{K} h(\boldsymbol{Z}_v) \boldsymbol{Z}_v^{(k)}\Big), \tag{12}$$

## 4.5 Expert-aware Dynamic Pseudo-labeling

For each unlabeled node $v$, we generate the sharpening pseudo-label $\widetilde{\boldsymbol{Y}}_v$ from the MoE framework, which is then used for the node classification. In particular, we use the $\theta$-percentile of the entropy value $u_v^{(k)}$ from each expert to establish a dynamic threshold $\tau$, which can be defined as:

$$\tau = \text{Percentile}_\theta \Big(\{u_v^{(k)}\}_{k=1}^{K}\Big). \tag{13}$$

Given this dynamic threshold, we filter out noisy pseudo labels and leave only the high-quality samples from the unlabeled data. In formula:

$$\mathcal{V}_{\text{selected}} = \{(v, u_v^{(k^*)}) | u_v^{(k^*)} \leq \tau\}. \tag{14}$$

We leverage the high-quality samples to fine-tune the model, thereby improving its performance and promoting better generalisation under domain shifts. The unsupervised classification loss can be:

$$\mathcal{L}_u = \sum_{v \in \mathcal{V}_{\text{selected}}} \text{CE}\Big(\widetilde{\boldsymbol{Y}}_v, f\Big(\sum_{k=1}^{K} h(\boldsymbol{Z}_v) \boldsymbol{Z}_v^{(k)}\Big)\Big). \tag{15}$$

Table 1: The model performance comparison on six domain generalization tasks with source label rate as 5%. A: ACMv9; C:Citationv1; D: DBLPv7. A⇒C represents that A is the source graph and C is the target graph. The same applies to other tasks.

| Methods | A⇒C | | A⇒D | | C⇒A | | C⇒D | | D⇒A | | D⇒C | |
|---|---|---|---|---|---|---|---|---|---|---|---|---|
| | Micro | Macro | Micro | Macro | Micro | Macro | Micro | Macro | Micro | Macro | Micro | Macro |
| MLP | $41.3_{\pm1.15}$ | $35.8_{\pm0.72}$ | $42.8_{\pm0.88}$ | $36.3_{\pm0.77}$ | $39.4_{\pm0.57}$ | $33.7_{\pm0.58}$ | $43.7_{\pm0.69}$ | $36.7_{\pm0.55}$ | $37.3_{\pm0.32}$ | $30.8_{\pm0.37}$ | $39.4_{\pm0.99}$ | $32.8_{\pm0.99}$ |
| GCN | $54.4_{\pm1.52}$ | $52.0_{\pm1.62}$ | $56.9_{\pm2.33}$ | $53.4_{\pm2.81}$ | $54.1_{\pm1.40}$ | $52.3_{\pm1.98}$ | $58.9_{\pm0.99}$ | $54.5_{\pm1.55}$ | $50.1_{\pm2.14}$ | $48.0_{\pm3.28}$ | $56.0_{\pm1.24}$ | $51.9_{\pm1.49}$ |
| GSAGE | $49.3_{\pm2.18}$ | $46.4_{\pm2.06}$ | $51.8_{\pm1.35}$ | $47.4_{\pm1.62}$ | $46.8_{\pm2.56}$ | $45.0_{\pm2.78}$ | $51.7_{\pm1.95}$ | $48.1_{\pm1.97}$ | $41.7_{\pm2.17}$ | $37.4_{\pm4.59}$ | $45.4_{\pm2.11}$ | $39.3_{\pm3.45}$ |
| GAT | $55.1_{\pm3.22}$ | $50.8_{\pm1.45}$ | $55.3_{\pm2.52}$ | $51.8_{\pm2.60}$ | $50.0_{\pm1.20}$ | $45.6_{\pm2.36}$ | $55.4_{\pm2.73}$ | $49.2_{\pm2.59}$ | $44.8_{\pm2.74}$ | $38.3_{\pm4.84}$ | $50.4_{\pm3.35}$ | $42.0_{\pm4.46}$ |
| GIN | $64.6_{\pm2.47}$ | $56.0_{\pm2.73}$ | $60.0_{\pm2.09}$ | $51.3_{\pm3.99}$ | $57.1_{\pm1.19}$ | $54.4_{\pm2.57}$ | $62.0_{\pm1.05}$ | $56.8_{\pm1.40}$ | $51.9_{\pm2.00}$ | $45.4_{\pm2.16}$ | $60.2_{\pm3.05}$ | $53.0_{\pm2.10}$ |
| CaNet | $62.3_{\pm3.11}$ | $54.9_{\pm2.98}$ | $61.2_{\pm3.77}$ | $51.0_{\pm2.00}$ | $58.6_{\pm1.01}$ | $51.3_{\pm0.76}$ | $63.0_{\pm2.31}$ | $57.5_{\pm1.64}$ | $52.4_{\pm5.09}$ | $43.8_{\pm1.77}$ | $57.2_{\pm4.01}$ | $48.3_{\pm3.67}$ |
| MARIO | $65.5_{\pm4.07}$ | $60.9_{\pm4.31}$ | $66.3_{\pm5.85}$ | $61.4_{\pm4.05}$ | $57.0_{\pm4.16}$ | $53.5_{\pm5.10}$ | $63.9_{\pm4.89}$ | $53.7_{\pm5.56}$ | $51.5_{\pm4.13}$ | $42.9_{\pm3.06}$ | $63.9_{\pm5.28}$ | $55.7_{\pm4.57}$ |
| LDAT | $69.3_{\pm0.99}$ | $70.3_{\pm1.30}$ | $65.0_{\pm2.24}$ | $67.0_{\pm1.53}$ | $65.8_{\pm0.89}$ | $63.1_{\pm1.65}$ | $70.7_{\pm1.57}$ | $68.4_{\pm1.11}$ | $60.6_{\pm2.81}$ | $60.3_{\pm2.65}$ | $64.7_{\pm2.00}$ | $62.3_{\pm1.87}$ |
| SGDA | $71.8_{\pm1.85}$ | $70.0_{\pm1.83}$ | $63.3_{\pm3.01}$ | $61.6_{\pm2.15}$ | $66.2_{\pm1.60}$ | $65.3_{\pm1.74}$ | $68.5_{\pm1.40}$ | $65.8_{\pm1.29}$ | $56.9_{\pm1.01}$ | $57.4_{\pm0.73}$ | $62.3_{\pm0.77}$ | $61.8_{\pm0.85}$ |
| Ours | $\mathbf{75.7_{\pm0.89}}$ | $\mathbf{73.5_{\pm0.91}}$ | $\mathbf{70.7_{\pm0.50}}$ | $\mathbf{68.5_{\pm0.57}}$ | $\mathbf{70.6_{\pm0.51}}$ | $\mathbf{71.3_{\pm0.50}}$ | $\mathbf{73.3_{\pm0.58}}$ | $\mathbf{72.3_{\pm0.53}}$ | $\mathbf{64.5_{\pm0.38}}$ | $\mathbf{64.3_{\pm0.50}}$ | $\mathbf{73.6_{\pm0.61}}$ | $\mathbf{69.0_{\pm1.71}}$ |

## 4.6 SUMMARIZATION

To ensure each expert captures distinct aspects of the ego-subgraphs, we propose a diversity objective with distance correlation Székely et al. (2007); Wang et al. (2020) to encourage divergence:

$$\mathcal{L}_{ind} = \sum_{k=1}^{K} \sum_{k'=k+1}^{K} dCor(\mathbf{Z}^{(k)}, \mathbf{Z}^{(k')}), \quad (16)$$

where $\mathbf{Z}^{(k)}$ is the representation of all nodes and $dCor(\cdot)$ is the distance correlation function:

$$dCor(\mathbf{Z}^{(k)}, \mathbf{Z}^{(k')}) = \frac{dCov(\mathbf{Z}^{(k)}, \mathbf{Z}^{(k')})}{\sqrt{dVar(\mathbf{Z}^{(k)}) \cdot dVar(\mathbf{Z}^{(k')})}}, \quad (17)$$

where $dVar(\cdot)$ and $dCov(\cdot)$ is the distance variance and covariance respectively. Finally, we combine supervised classification loss $\mathcal{L}_s$ from labeled node, the knowledge distillation loss $\mathcal{L}_{dl}$ and the expert diversity regularization $\mathcal{L}_{ind}$. The overall objective of our LIGHT is:

$$\mathcal{L} = \mathcal{L}_s + \mathcal{L}_u + \lambda_{dl}\mathcal{L}_{dl} + \lambda_{ind}\mathcal{L}_{ind}, \quad (18)$$

where $\lambda_{dl}$ and $\lambda_{ind}$ are the balance factors that control the weights of the losses. The overall algorithm of our proposed LIGHT can be found in the Appendix B.

## 5 EXPERIMENT

### 5.1 SETUP

**Datasets.** We conduct extensive experiments from ArnetMiner Tang et al. (2008), i.e., ACMv9, Citationv1, and DBLPv7. These datasets are widely used for evaluating graph learning models, where each paper, represented as a node, is to be categorized into one of five predefined research areas, including Artificial Intelligence, Computer Vision, Database, Information Security, and Networking. The citation relationship is represented as an edge. These datasets provide diverse graph structures and data contributions, making them suitable to evaluate our method.

**Compared Baselines.** To comprehensively evaluate our proposed LIGHT, we compare it against a comprehensive suite of baseline methods including GCN Kipf & Welling (2017), GraphSAGE (GSAGE) Hamilton et al. (2017), GAT Vrahatis et al. (2024), GIN Xu et al. (2019), CaNet Wu et al. (2024a), MARIO Zhu et al. (2024), LDAT Wang et al. (2022), and SGDA Qiao et al. (2023a). Note that we adopt the same experimental settings with baselines to ensure a fair comparison.

**Implementational Details.** All experiments are conducted on a single NVIDIA RTX3090 GPU with 24GB memory. We utilize PyTorch and PyG to implement our framework. We utilize a random splitting strategy to ensure diversity. In the source domain, 5% of node labels are used for supervised training. The MoE network is composed of multiple experts and a router network. All experiments are conducted with five fixed random seeds for reproducibility. Our LIGHT is trained using the Adam optimizer with a learning rate of 1e-3 and L2 weight decay of 2e-3. We utilize Qwen2.5-7B as our default LLMs and attempt the Gemma3 series model in our experiments as well. We report the performance on the target domain nodes in terms of both Micro F1-score and Macro F1-score.

Table 2: Ablation studies on the Citations benchmark (MoE/LLM/DIV/HQ as peers).

| Models | A ⇒ C | | A ⇒ D | | C ⇒ A | | C ⇒ D | | D ⇒ A | | D ⇒ C | |
|---|---|---|---|---|---|---|---|---|---|---|---|---|
| | Micro | Macro | Micro | Macro | Micro | Macro | Micro | Macro | Micro | Macro | Micro | Macro |
| LIGHT w/o MoE | 43.3 | 37.1 | 44.5 | 38.5 | 42.1 | 34.3 | 46.8 | 37.8 | 36.0 | 31.5 | 38.2 | 33.9 |
| LIGHT w/o LLM | 72.0 | 66.6 | 66.7 | 58.5 | 66.1 | 63.3 | 69.7 | 64.7 | 51.0 | 43.4 | 66.0 | 61.7 |
| LIGHT w/o DIV | 74.9 | 72.7 | 68.6 | 66.9 | 67.5 | 68.1 | 71.8 | 71.2 | 60.3 | 57.5 | 68.0 | 61.8 |
| LIGHT w/o HQ | 70.0 | 66.0 | 67.5 | 62.2 | 64.4 | 60.4 | 71.1 | 69.2 | 57.5 | 52.0 | 57.2 | 50.1 |
| LIGHT | **75.7** | **73.5** | **70.7** | **68.5** | **70.6** | **71.3** | **73.3** | **72.3** | **64.5** | **64.3** | **73.6** | **71.6** |

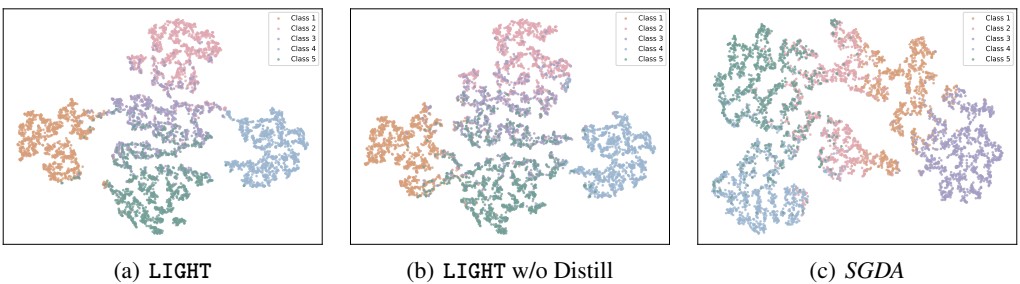

(a) LIGHT            (b) LIGHT w/o Distill            (c) *SGDA*

Figure 2: Analysis of different t-SNE visualizations on $A \Rightarrow C$ task. (a), (b) respectively represent the performance of our method before and after knowledge distillation, (c) represents the effect of the baseline method SGDA.

## 5.2 EMPIRICAL PERFORMANCE

**Performance Comparison.** The compared results can be found in Table 1, across all six domain generalization tasks involving transfers between ACMv9 (A), Citationv1 (C), and DBLPv7 (D) datasets, even with a scarce 5% source label rate. ▷ *Firstly*, our LIGHT consistently achieves state-of-the-art Micro-F1 and Macro-F1 scores. For instance, on the A⇒C task, our method achieves 75.7% Micro-F1, surpassing the strongest specialized baseline SGDA (71.8%) by 3.9% and exhibiting larger improvements over standard GNNs like GIN. The performance increasement can be attributed to two factors as follows: (I) Introduction of our LLM-based guidance, which provides the judgments of the most graph experts for strong generalizability with knowledge distillation. (II) Introduction of our mixture-of-experts framework, which provides adaptive neighborhood aggregation for topological exploration. ▷ *Secondly*, LLM guidance dynamically selects experts, especially for nodes with high uncertainty, providing an adaptability which absent in baselines. This tailored expertise, combined with a semi-supervised learning component that leverages high-quality pseudo-labels identified through a novel process considering both expert agreement and LLM-derived confidence, facilitates effective knowledge distillation and transfer from limited labeled data. The consistent high performance across all tasks, which involve datasets with varied structural properties and data distributions, underscores the robustness and generalizability of LIGHT.

**Ablation Study.** To validate the individual contributions of key components in our LIGHT, we introduce the following model variants: (I) LIGHT w/o MoE, which removes the MoE framework and utilize one single expert; (II) LIGHT w/o LLM, which removes the knowledge distillion from LLMs; (III) LIGHT w/o DIV, which removes the divergence-prompting objective in our framework; (IV) LIGHT w/o HQ, which removes our expert-aware dynamic pseudo-labeling. Results are shown in Table 2. We can observed that removing the MoE framework led to the most substantial performance decline, underscoring its fundamental role. Disabling LLM-guided expert selection caused Micro-F1 scores decreasing by about 7.3% and Macro-F1 by 12.9%, highlighting the LLM's contribution to refining expert routing. The exclusion of the high-quality pseudo-labeling strategy resulted in a considerable reduction in performance, averaging a 7.0% decrease in Micro-F1 and an 11.6% decrease in Macro-F1, emphasizing its importance in leveraging unlabeled data. Finally, removing the MoE's diversity/coherence loss led to Micro-F1 scores dropping by an average of 2.9% and Macro-F1 by 4.0%, indicating its value in refining expert representations. These findings collectively affirm the crucial role each component plays in the proposed method's overall effectiveness.

**Case Study.** To further investigate the qualitative impact of our proposed method, particularly the LLM-guided expert selection distillation, we present a case study visualizing the learned node embeddings from a representative domain generalization task (e.g., A⇒C). The t-SNE van der Maaten

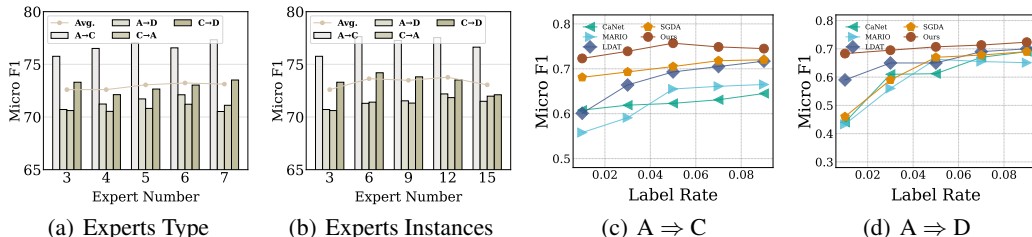

(a) Experts Type     (b) Experts Instances     (c) A ⇒ C     (d) A ⇒ D

Figure 3: (a) shows the Micro F1 score for varying expert types, and (b) illustrates results when expert instances increase proportionally. Sub-figures (c) and (d) compare performance under different label rates on A ⇒ C and A ⇒ D tasks respectively.

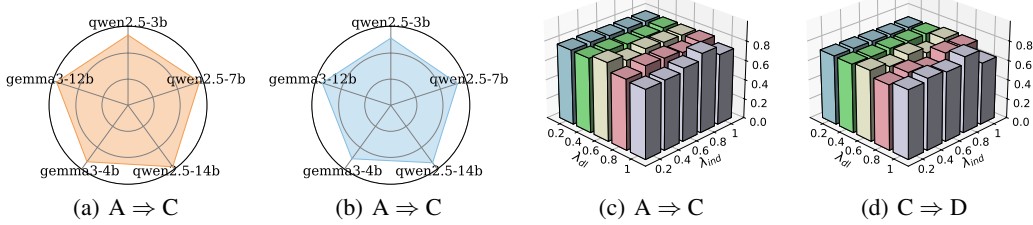

(a) A ⇒ C     (b) A ⇒ C     (c) A ⇒ C     (d) C ⇒ D

Figure 4: Effect analysis of LLMs type and loss weight effect impact analyze on micro-f1 score.

& Hinton (2008) results in Figure 2 illustrates these visualizations for three distinct model configurations: (a) our full proposed method including LLM distillation (ours w distill), (b) an ablation of our method without the LLM distillation component (ours w/o distill), and (c) the SGDA baseline, a relevant graph domain adaptation method. Node colors correspond to their ground-truth classes. By providing high-level, context-aware guidance to the MoE's gating mechanism, the LLM helps the model assign nodes to the most appropriate specialized experts, leading to embeddings that better align with the true class distinctions. The noticeable degradation in cluster quality when distillation is removed corroborates this, further emphasizing the value added by the LLM component.

**Parameter Analysis.** In this part, we conduct parameter analysis. The number of experts influences performance in Figure 3 (a), where increasing the diversity of expert types or the number of instances per expert type in Figure 3 (b) can yield improvements, though gains diminish beyond an optimal point. Regarding data scarcity, while higher label rates generally lead to better outcomes, the proposed method demonstrates strong performance even with low label rates, outperforming alternatives in semi-supervised domain generalization. The results in Figure 4 (c) and (d) shows the choice of LLM for guidance affects outcomes, with larger or more capable LLMs generally offering better guidance, although the performance gap between mid-sized (e.g., 7b) and larger (e.g., 12-14b) models was not substantial, while smaller models (e.g., 3-4b) showed a noticeable performance decrease. Further analysis is provided in the Appendix F. Finally, the model's performance is sensitive to the loss weight hyperparameters ($\lambda_{dl}$, $\lambda_{ind}$), which control the balance between unsupervised classification, knowledge distillation, and expert diversity regularization losses; their optimal values vary across some tasks, tuning as illustrated by the results in Figure 3 (c) and (d).

## 6 CONCLUSION

In this work, we have introduced a novel framework `LIGHT` for semi-supervised domain generalization on graphs. Our `LIGHT` leverages the zero-shot ability of LLMs to steer a MoE architecture, where each expert specializes in different graph topological neighborhoods. This allows for dynamic, context-aware selection of the most pertinent expert, guided by multi-hop ego-subgraph prompts transformed for LLM comprehension. Furthermore, we have incorporated an expert-enhanced pseudo-labeling strategy, which leverages high-confidence pseudo-labels to effectively supervise the model's training in label-scarce environments. Extensive empirical evaluations on multiple public benchmark datasets demonstrate the superiority of our method in generalizing across different graph domains, significantly outperforming existing state-of-the-art methods.

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

## A  LARGE LANGUAGE MODEL (LLM) USAGE STATEMENT

We use the LLM as a general-purpose assistant tool. Specifically, the LLM assists in (i) checking grammar and improving clarity of text descriptions, and (ii) suggesting alternative phrasings for some sections. No parts of the paper are generated entirely by the LLM. All research ideas, experiments, model designs, and results are conceived, implemented, and analyzed solely by the authors. The LLM does not contribute to the development of the methodology, experiments, or analysis presented in this paper. We confirm that the use of the LLM is limited to minor writing support and does not constitute a substantive contribution that would qualify it as a co-author.

## B  THEORETICAL ANALYSIS

In this section, we present a theoretical analysis of the proposed LIGHT framework. We assume the existence of a set of $K$ experts, each specialized in learning a particular domain. Each domain may be subject to potential distribution shifts, making domain-specific modeling essential. For notational simplicity, we focus on a binary classification setting with $C = 2$ classes and assume the presence of two distinct domains. Suppose that for the first $K/2$ experts, the domain-specific distribution is aligned with the first domain, while the remaining $K/2$ experts are aligned with the second domain. Assume that the routing weights $\boldsymbol{\omega}_v^{(k)}$ satisfy the following condition: for the correct expert $k$,

$$\boldsymbol{\omega}_v^{(k)} = \frac{2}{K}, \quad \text{and for all incorrect experts } k' \neq k, \quad \boldsymbol{\omega}_v^{(k')} = 0,$$

which indicates that the routing mechanism assigns equal weight to the correct expert while ignoring all others. Let the final prediction for node $v$ be defined as:

$$\hat{\boldsymbol{Y}}_v = f\left(\sum_{k=1}^{K} \boldsymbol{\omega}_v^{(k)} \boldsymbol{Z}_v^{(k)}\right),$$

where $\boldsymbol{Z}_v^{(k)}$ is the representation of node $v$ obtained from the $k$-th expert, $\boldsymbol{\omega}_v^{(k)}$ denotes the routing weight for expert $k$, and $f(\cdot)$ is a learnable classifier. We further assume that the classifier $f(\cdot)$ can be expressed as a logistic (sigmoid) transformation of a linear combination of expert outputs:

$$f\left(\sum_{k=1}^{K} \boldsymbol{\omega}_v^{(k)} \boldsymbol{Z}_v^{(k)}\right) = \sigma\left(\sum_{k=1}^{K} \boldsymbol{\omega}_v^{(k)} \boldsymbol{W}^\top \boldsymbol{Z}_v^{(k)}\right),$$

where $\sigma(\cdot)$ is the sigmoid activation function and $\boldsymbol{W}$ is the weight matrix of the classifier. Under this formulation, the term $\boldsymbol{W}^\top \boldsymbol{Z}_v^{(k)}$ can be interpreted as the logit output of the $k$-th expert for node $v$. Due to the monotonicity of the sigmoid function, a larger logit corresponds to a higher predicted probability of class 1. The following lemma establishes a threshold-based decision rule for the aggregated expert logits.

**Lemma B.1.** *There exists a threshold $\tau \in \mathbb{R}$ such that:*

$$\hat{\boldsymbol{Y}}_v = \begin{cases} 1, & \text{if } \sum_{k=1}^{K} \boldsymbol{\omega}_v^{(k)} \boldsymbol{W}^\top \boldsymbol{Z}_v^{(k)} > \tau, \\ 0, & \text{otherwise.} \end{cases}$$

In the multi-class setting ($C > 2$), a similar formulation applies by replacing the sigmoid function with the softmax function, and defining a class-specific threshold for each output dimension.

With this lemma, we can analyze the performance of the proposed LIGHT framework. Specifically, we aim to show that a mixture-of-experts (MoE) model can outperform a single-expert model in terms of classification accuracy. Without loss of generality, we assume that the threshold $\tau$ in Lemma B.1 is set to 0. Let the logit output of the $k$-th expert for node $v$ be denoted as

$$Q_{v,k} = \boldsymbol{W}^\top \boldsymbol{Z}_v^{(k)},$$

and let its expected value be $\mu_{v,k} = \mathbb{E}[Q_{v,k}]$. For a given node $v$, suppose expert $k$ is the correct expert (i.e., best aligned with the domain-specific distribution of $v$). Then, the expected logit $\mu_{v,k}$ should exceed a positive margin $\nu > 0$, while any incorrect expert $k' \neq k$ should have mean

logit output $\mu_{v,k'} = 0$. This reflects the fact that only the correct expert can effectively model the distributional characteristics of node $v$. We further assume that the logits of different experts are independent and sub-Gaussian distributed, i.e.,

$$\mathbb{E}[\exp t(Q_{v,k} - \mu_{v,k})] \leq \exp \frac{t^2 \sigma^2}{2}, \quad \forall t \in \mathbb{R},$$

where $\sigma^2$ is a constant. With these assumptions, we can analyze the prediction error of the MoE model.

*Proof of Theorem 4.1.* We begin by analyzing the case where only a single expert is available, i.e., $K = 1$. Suppose the true label for node $v$ is 1. The prediction error of the model is then given by

$$\mathbb{P}(\hat{\boldsymbol{Y}}_v \neq 1) = \mathbb{P}(Q_{v,1} < 0).$$

We aim to show that this probability is strictly greater than zero, i.e., $\mathbb{P}(Q_{v,1} < 0) > 0$.

Since the expert is misaligned with the domain-specific distribution of node $v$, we assume $\mathbb{E}[Q_{v,1}] = 0$. Define the following events:

$$\mathcal{A}_1 = \{Q_{v,1} < 0\}, \quad \mathcal{A}_2 = \{Q_{v,1} \geq 0\}, \quad \mathcal{A}_3 = \{Q_{v,1} = 0\}.$$

Because $Q_{v,1}$ is not almost surely constant, we must have $\mathbb{P}(\mathcal{A}_3) < 1$. If $\mathbb{P}(Q_{v,1} < 0) = 0$, then necessarily $\mathbb{P}(Q_{v,1} > 0) > 0$, which contradicts the assumption that $\mathbb{E}[Q_{v,1}] = 0$. Therefore, it must be that $\mathbb{P}(Q_{v,1} < 0) > 0$, and the prediction error is strictly positive in the single-expert case.

Now consider the case with $K$ experts. Suppose expert $k$ is correctly aligned with the domain of node $v$, and thus has expected logit output $\mu_{v,k} = \nu > 0$, while all other experts $k' \neq k$ have zero-mean outputs, $\mu_{v,k'} = 0$. Then the expected value of the aggregated logit is

$$\mathbb{E}\left[\sum_{k=1}^{K} \boldsymbol{\omega}_v^{(k)} Q_{v,k}\right] = \sum_{k=1}^{K} \boldsymbol{\omega}_v^{(k)} \mu_{v,k} = \nu.$$

Assuming the expert outputs $Q_{v,k}$ are independent and sub-Gaussian with variance proxy $\sigma^2$, we apply Hoeffding's inequality to bound the tail probability:

$$\mathbb{P}\left(\sum_{k=1}^{K} \boldsymbol{\omega}_v^{(k)} Q_{v,k} < 0\right) \leq \exp\left(-\frac{C\nu^2}{2\sigma^2/K}\right).$$

Therefore, the prediction error of the MoE model satisfies:

$$\mathbb{P}(\hat{\boldsymbol{Y}}_v \neq 1) \leq \exp\left(-\frac{C\nu^2 K}{2\sigma^2}\right) \leq \frac{C_0}{K},$$

for some constant $C_0 > 0$. This completes the proof. $\square$

## C  GREEDY ANCHOR NODE SELECTION

---

**Algorithm 1** Greedy Anchor Node Selection

---

**Input**: PPMI matrix $\boldsymbol{P}$, Influence Score $I_v(u)$ from node $v$ to node $u$, budget $B$.
**Output**: Anchor Node set $S$

---

1. $S = \emptyset$
2. **for** $t = 1, 2, \ldots, B$ **do**
    (a) **for** $v \in \mathcal{V}_{train} \setminus S$ **do**
        i. update $\sigma(S \cup \{v\})$ based on Eq. (9)
    (b) **end for**
    (c) $v^* = \underset{v \in \mathcal{V}_{train} \setminus S}{\arg\max} \left(|\sigma(S \cup \{v\})| - |\sigma(S)|\right)$
    (d) $S = S \cup \{v^*\}$
3. **end for**
4. **return** $S$

---

# D    BASELINE DETAILS

To comprehensively evaluate our method, we compare it against a comprehensive suite of baseline methods. These baselines are carefully selected to cover a range of approaches: (i) a fundamental neural network model that processes node features independently of graph structure; (ii) standard Graph Neural Network architectures trained via empirical risk minimization on the source domain, representing common graph representation learning techniques without explicit domain generalization mechanisms; and (iii) contemporary methods specifically designed for graph out-of-distribution generalization or domain adaptation, which employ diverse strategies to tackle distribution shifts. Following previous works Kipf & Welling (2017); Qiao et al. (2023a), all GNN-based baselines are trained using the 5% labeled data from the source domain for the specified cross-domain generalization tasks. To assess performance against established graph representation learning techniques, we include several widely-adopted architectures. We begin with an MLP which serves to quantify the performance achievable without considering the intrinsic graph structure.

# E    FURTHER ANALYSIS OF ABLATION EXPERIMENT

To meticulously validate the individual contributions of the key components within our proposed method, we conducted a series of ablation experiments. We systematically removed or simplified core modules: (1) the Mixture of Experts architecture (w/o MoE), reverting to a more standard single MLP encoder. Without the expert mechanism, all other modules based on it are also removed; (2) the LLM-guided expert selection distillation (w/o LLM); (3) the MoE's internal diversity/coherence loss (w/o DIV); and (4) our expert-aware dynamic pseudo-labeling strategy for semi-supervised learning (w/o HQ). The performance of these variants was evaluated on the same six domain generalization tasks, with results and the extent of performance degradation detailed in Table 3.

Table 3: Ablation studies on the Citations benchmark with component presence indicators.

| Components | | | | A ⇒ C | | A ⇒ D | | C ⇒ A | | C ⇒ D | | D ⇒ A | | D ⇒ C | |
|---|---|---|---|---|---|---|---|---|---|---|---|---|---|---|---|
| PPMI | LLM | DIV | HQ | Micro | Macro | Micro | Macro | Micro | Macro | Micro | Macro | Micro | Macro | Micro | Macro |
| | | | ✓ | 71.3 | 66.0 | 65.9 | 58.8 | 65.7 | 63.1 | 69.1 | 63.6 | 50.4 | 42.6 | 65.7 | 61.3 |
| | ✓ | | | 68.8 | 64.4 | 65.7 | 60.1 | 62.8 | 58.7 | 68.2 | 67.3 | 55.8 | 49.7 | 54.6 | 49.2 |
| | ✓ | ✓ | | 69.6 | 65.3 | 67.0 | 62.0 | 63.7 | 60.2 | 71.0 | 68.7 | 56.7 | 51.2 | 56.4 | 49.9 |
| | ✓ | | ✓ | 73.2 | 72.0 | 68.2 | 66.1 | 66.7 | 67.4 | 70.4 | 70.0 | 58.9 | 55.1 | 66.1 | 59.8 |
| ✓ | | ✓ | | 72.0 | 66.6 | 66.7 | 58.5 | 66.1 | 63.3 | 69.7 | 64.7 | 51.0 | 43.4 | 66.0 | 61.7 |
| ✓ | ✓ | | | 69.3 | 65.7 | 66.2 | 61.0 | 63.1 | 59.1 | 69.8 | 68.5 | 56.3 | 50.7 | 55.0 | 49.9 |
| ✓ | ✓ | ✓ | | 70.0 | 66.0 | 67.5 | 62.2 | 64.4 | 60.4 | 71.1 | 69.2 | 57.5 | 52.0 | 57.2 | 50.1 |
| ✓ | ✓ | | ✓ | 74.9 | 72.7 | 68.6 | 66.9 | 67.5 | 68.1 | 71.8 | 71.2 | 60.3 | 57.5 | 68.0 | 61.8 |
| ✓ | ✓ | ✓ | ✓ | **75.7** | **73.5** | **70.7** | **68.5** | **70.6** | **71.3** | **73.3** | **72.3** | **64.5** | **64.3** | **73.6** | **71.6** |

**Impact of MoE Architecture (MoE).** The results show removing the Mixture of Experts framework leads to the most substantial performance decline across all tasks and metrics, with Micro-F1 scores dropping by an average of approximately 29.2% and Macro-F1 by 34.5%. For instance, on A⇒C, the Micro-F1 score plummets by 32.4 points. Notice without MoE Architecture, all other experts related mechanism are disabled and our framework revert to simple MLP-based method. This drastically reduced performance underscores that the ability of MoE to employ specialized experts and adaptively combine their insights is fundamental to our model's success, exceeding the capabilities of a monolithic encoder structure in these challenging domain generalization scenarios.

**Impact of LLM-guided Expert Selection (LLM).** Disabling the LLM-based distillation for expert selection results in a significant performance drop, with Micro-F1 scores decreasing by an average of approximately 7.3% and Macro-F1 by 12.9%. The degradation is particularly notable on tasks like D⇒A (13.5 points drop in Micro-F1). This highlights the contribution of the LLM's semantic guidance in refining the MoE's expert routing, especially for uncertain nodes. Without this external, high-level reasoning, the MoE's autonomous gating mechanism is demonstrably less effective at tailoring expertise to specific node contexts for optimal generalization.

**Impact of MoE Diversity/Coherence Loss (DIV).** Removing the MoE's internal diversity loss results in a noticeable, albeit comparatively smaller, performance decrease, with Micro-F1 scores dropping by an average of 2.9% and Macro-F1 by 4.0%. While more modest, this degradation across tasks indicates that this regularization term plays a valuable role in refining the representations learned by each expert. By encouraging intra-expert coherence or better-defined specializations, the diverse neighborhood exploration also avoid the routing network get into local optimal solution.

**Impact of Expert-aware Dynamic Pseudo-labeling (HQ).** The exclusion of our specialized expert-aware dynamic pseudo-labeling strategy also leads to a considerable reduction in performance, averaging a 7.0% decrease in Micro-F1 and an 11.6% decrease in Macro-F1. This emphasizes the importance of this sophisticated semi-supervised learning component, which selectively leverages unlabeled source data. By identifying reliable pseudo-labels through a process informed by expert consistency and LLM insights, our method effectively learns from the scarce 5% labeled data and mitigates error propagation common in naive pseudo-labeling approaches.

**Impact of Positive Pointwise Mutual Information (PPMI).** We replace $P_v^{(k)}$ to the original graph adjacency matrix. This results of the overall performance has decreased by about 1-2%. The reason is that without such a model architecture highly sensitive to topological structures, the differentiation in feature extraction among heterogeneous experts diminishes. Consequently, it becomes more challenging for the router to assign nodes to appropriate experts.

## F  FURTHER ANALYSIS OF PARAMETER SENSITIVITY

**Number of Experts.** We further investigated the influence of the number of experts on the model's performance. Figure 3 (a) and (b) illustrates two scenarios: *Varying Expert Types*: The results from (a) shows how performance changes as the diversity and numbers of expert types increases. The results suggest that increasing the variety of specialized experts can merely positively impact the Micro F1 score up to a certain point by providing more capacity or stability to the specialized processing. *Varying Expert Instances*: The results from (b) demonstrates the effect of increasing the number of instances for each type of expert. This indicates that having more instances of the same expert type can also contribute to performance improvements, after which the gains may diminish or performance could even degrade.

**Label Rate.** Figure 3 (c) and (d) examines our model's performance and robustness to the scarcity of labeled data by varying the label rates in the source domain. The experiments were conducted on the $A \Rightarrow C$ and $A \Rightarrow D$ tasks. As expected, a higher label rate generally leads to better performance. However, the proposed method demonstrates strong performance with low label rates, outperforming other compared methods, highlighting its effectiveness in semi-supervised domain generalization scenarios.

**Loss Weights.** The impact of different loss weight hyperparameters ($\lambda_u$, $\lambda_{dl}$, $\lambda_{ind}$) on the Micro-F1 score was analyzed, as shown in the heatmaps in Figure 4 (c) and (d) for tasks $A \Rightarrow C$, $C \Rightarrow D$. These hyperparameters control the balance between the unsupervised classification loss, the knowledge distillation loss, and the expert diversity regularization loss, respectively. The results indicate that the optimal weights can vary depending on the specific domain generalization task. Generally, the model's performance is sensitive to these weights, and careful tuning is required to achieve the best results. For instance, in the $A \Rightarrow C$ task, specific ranges for $\lambda_u$, $\lambda_{dl}$, and $\lambda_{ind}$ yield higher Micro-F1 scores, as depicted by the color intensity in the heatmap. Similar trends, though with potentially different optimal regions, are observed for the $C \Rightarrow D$ and $D \Rightarrow C$ tasks.

**Smoothing Coefficient and Activation Threshold.** Figure 5 presents an analysis of the smoothing coefficient ($\alpha$) used in the label smoothing strategy for expert selection and the activation threshold ($\theta$) used in the expert-enhanced pseudo-labeling process. These experiments were performed on the $A \Rightarrow C$, $A \Rightarrow D$, $C \Rightarrow A$ and $C \Rightarrow D$ tasks. The smoothing coefficient $\alpha$ determines the softness of the target distribution provided by the LLM for training the routing function. The activation threshold $\theta$ is used to filter noisy pseudo-labels by selecting high-confidence samples. The results suggest that model performance is influenced by the interplay of these two parameters. Most of optimal performance is achieved within specific ranges of $\alpha = 0.8$ and $\theta = 0.05$, indicating that both appropriate smoothing of LLM guidance and careful selection of pseudo-labels are important for the model's success.

**LLM Types.** We further explored the effect of using different types of LLMs for the LLM-guided expert selection, as illustrated in Figure 4 (a) and (b) for the $A \Rightarrow C$ and $A \Rightarrow D$ tasks. The results compare the performance achieved using various LLMs, such as qwen2.5-3b, qwen2.5-7b, qwen2.5-14b Qwen et al. (2025), gemma3-4b, and gemma3-12b Team et al. (2025). The results indicate that the choice of LLM can impact the model's performance. Larger or more capable LLMs (e.g., qwen2.5-14b or gemma-12b in some cases) tend to provide better guidance, leading to

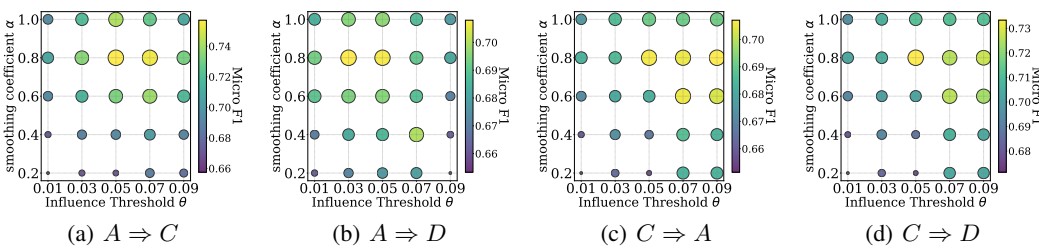

(a) $A \Rightarrow C$          (b) $A \Rightarrow D$          (c) $C \Rightarrow A$          (d) $C \Rightarrow D$

Figure 5: Effect analysis of smoothing coefficient $\alpha$ and influence threshold $\theta$.

improved results, though the best-performing LLM might vary depending on the specific task. We observe 3-4% of performance degradation with small size model like 3-4b, while the performance gap between the 7b model and 12-14b model is not significant.

## G    LIMITATIONS

While the proposed framework demonstrates promising results in semi-supervised domain generalization on graphs, we acknowledge certain limitations that offer avenues for future research.

Firstly, our empirical evaluation was conducted on established benchmark datasets in academic research. Although these datasets are standard for assessing graph learning models, the performance of `LIGHT` on graphs with substantially different characteristics, such as highly dynamic graphs or graphs from vastly different domains (e.g., biological networks or social networks with unique structures), may vary. Further investigation across a broader spectrum of graph types and real-world applications would be beneficial to ascertain the generalizability of our approach.

Secondly, the use of LLMs and graph data inherently touches upon broader considerations of privacy and fairness. Depending on the nature of the graph data and the information processed by the LLM, potential privacy implications could arise. Similarly, fairness in node classification could be affected by biases present in either the graph data or the LLM. Future research could explicitly address these aspects by incorporating privacy-preserving techniques or fairness-aware learning objectives within our framework.

Addressing these limitations could further enhance the robustness, applicability, and societal alignment of the `LIGHT` framework and similar approaches leveraging LLMs for graph-based machine learning.

## H    SOCIETAL IMPACTS

Our method primarily focuses on advancing graph neural networks for semi-supervised domain generalization, aiming to improve their performance on unseen graphs with limited labeled data. The core idea involves using Large Language Models to guide a mixture-of-experts framework, enhancing knowledge distillation and addressing label scarcity through dynamic pseudo-labeling. The successful implementation of this approach could lead to:

**Improved Machine Learning Model Generalizability**: The LIGHT framework's ability to generalize effectively to unseen domains, even with scarce labeled data, can lead to more robust and adaptable machine learning models across various applications. This is particularly relevant in fields where data distribution shifts are common and obtaining labeled data is challenging or expensive.

**Enhanced Performance in Various Domains**: The paper highlights node classification as a fundamental task with wide applications in areas such as social network analysis, biochemistry, and recommender systems. Improvements in this area, as demonstrated by the LIGHT framework, could translate to more accurate predictions and better decision-making in these domains.

**Advancements in Semi-supervised Learning**: The research contributes to the field of semi-supervised learning on graphs by pioneering the use of LLM zero-shot capabilities to address domain

generalization under label scarcity. This could inspire new methodologies for leveraging LLMs in data-efficient graph learning.

**Better Understanding of Complex Relational Data**: By effectively capturing intricate relational and structural dependencies in graph data, this research can contribute to a deeper understanding of complex systems and networks.

We also acknowledge Potential Negative Societal Impacts:

**Potential for Misuse**: As with many advancements in machine learning, there's a general risk that improvements in GNNs and LLMs could be adapted for malicious purposes if not carefully governed.

**Fairness Considerations**: Biases present in either the graph data or the LLM could affect fairness in node classification. If the training data or the LLM itself contains biases against certain groups, the deployed technology could make decisions that unfairly impact these groups.

