# OpenReview forum: "LIGHT: LLM-guided Graph Expert Routing for Semi-supervised Domain Generalization"
_ICLR.cc/2026/Conference — ICLR 2026 Conference Withdrawn Submission_

### Official Review · Reviewer_eKzs · 2025-10-29

**Soundness:** 2
**Presentation:** 2
**Contribution:** 2
**Rating:** 2
**Confidence:** 4

**Summary:**

The paper introduces LIGHT, an LLM-Guided Graph Mixture-of-Experts framework for semi-supervised domain generalization on graphs. It aims to leverage large language models (LLMs) as a “judge” to route node representations to the most suitable graph expert based on multi-hop ego-subgraph prompts, and further refines training through expert-aware pseudo-labeling and knowledge distillation.

**Strengths:**

1. Addresses a timely problem: improving GNN generalization under label scarcity and distribution shift.

2. The overall framework is conceptually interesting: combining MoE routing with LLM guidance is interesting in principle.

3. Experimental results on several citation network datasets show consistent quantitative improvements.

**Weaknesses:**

1. The claimed task (“semi-supervised domain generalization on graphs”) is not clearly distinct from existing semi-supervised domain adaptation or transductive GSSL settings. The contribution boundary is vague.

2. Theorem 4.1 is a generic statement about MoE error reduction, offering no new insight or connection to LLM routing or domain shift.

3. The use of “LLM-as-a-judge” is heuristic and lacks justification. No evidence is provided that LLM reasoning truly improves invariant substructure selection compared to a learnable gating network.

4. The process of converting multi-hop subgraphs into text prompts is under-specified and likely unstable. Prompt wording or LLM choice could drastically change routing results.

5. All experiments are on closely related academic citation graphs (ACMv9, Citationv1, DBLPv7). It remains unclear whether the method generalizes to domains with different semantics or topology.

BTW: all the citation formation is wrong. You should use \cite or \citep, instead of \citet.

**Questions:**

1. What exactly differentiates “semi-supervised domain generalization” from semi-supervised domain adaptation or label-scarce transfer learning?

2. How sensitive is the routing to LLM model choice, prompt wording, or random seeds?

3. Can a purely learnable gating network (without LLMs) achieve similar performance?

4. What is the computational cost (in FLOPs / GPU hours / LLM tokens) compared to baselines?

5. How is Theorem 4.1 relevant to the overall method? What assumptions justify its bound?

6. Have you tested the framework on non-citation graphs (e.g., biological, social, or e-commerce networks) to verify generality?

**Details Of Ethics Concerns:**

The paper shows signs of LLM-generated or heavily templated writing (repetitive phrasing, generic sentences, weak logical transitions), which raises concerns about authorship transparency. I don't agree with what the authors claimed " No parts of the paper are generated entirely by the LLM. All research ideas, experiments, model designs, and results are conceived, implemented, and analyzed solely by the authors."

---

### Official Review · Reviewer_Mqoh · 2025-10-31

**Soundness:** 2
**Presentation:** 3
**Contribution:** 2
**Rating:** 4
**Confidence:** 5

**Summary:**

This paper presents LIGHT, a MoE-based framework for OOD generalization on graphs. Each GNN expert first produces prediction based on a designated hop of neighbors. Then a pretrained LLM serve as a routing module to determine the most suitable expert’s prediction. To mitigate label scarcity, LIGHT dynamically assigns pseudo-labels guided by expert entropy.

**Strengths:**

- The paper tackles a practically relevant yet challenging task.
- LIGHT demonstrates strong empirical performance with comprehensive quantitative analysis.

**Weaknesses:**

- The authors claim that directly using the LLM’s raw predictions leads to unreliable outputs. This makes me wonder whether the proposed method can overcome such issue compared to using LLM as s predictor. If the LLM’s reasoning is inherently unstable or prone to errors, then its guidance can still introduce noise regardless of its distillation format.
- The LLM-based routing still faces scalability issue as the number of neighbors increases, as the information of all neighbors must be encoded within a single prompt.
- The applicability of LIGHT may be constrained in graphs lacking textual attributes, which are frequently encountered in real-world scenarios. Moreover, its performance likely depends on the quality and richness of text descriptions. For instance, the LLM may struggle to make accurate/reliable judgements if given text attributes are too vague or short.
- LIGHT enforces a rigid hop-wise decomposition, which lacks fine-grained aggregation of neighborhood information. Often, the helpful signals for classification can be scattered *across* hops. However, the proposed rigid partitioning and cannot flexibly emphasize or suppress distinct contribution of neighbors beyond the hop-level.

**Questions:**

- What are the text attributes of nodes in benchmark graphs exactly? Are they paper abstracts?
- The current benchmarks are restricted to citation graphs. Could the authors evaluate the applicability of LIGHT into different types of text-attributed graphs?

---

### Official Review · Reviewer_1qUe · 2025-11-01

**Soundness:** 3
**Presentation:** 3
**Contribution:** 3
**Rating:** 4
**Confidence:** 3

**Summary:**

This paper proposes LIGHT, a LLM-guided MoE framework for semi-supervised domain generalization on graphs.

GNNs assume that training and testing data follow the same distribution (i.i.d.), but in reality, 'Out-of-Distribution (OOD)' situations, where the distribution differs, frequently occur, leading to significant performance degradation. Although various methodologies have been proposed to solve this, they have limitations due to problems such as (1) the problem of label scarcity and (2) the heterogeneity and simplicity bias of graph data.

To address this, LIGHT introduces three main components:
* Multi-hop Graph MoE (Mixture-of-Experts): This uses multiple 'expert' GNNs that explore neighborhood information at various depths, such as 0-hop, 1-hop, and 2-hop. This is to effectively respond to diverse data distributions and heterogeneous graph structures.
* LLM-based Context-aware Routing and Knowledge Distillation: The multi-hop neighborhood context of each node is transformed into a prompt and fed into the LLM. The LLM acts as a 'judge' rather than a 'predictor', guiding the selection of the most reliable expert for that node. Afterward, this judgment capability of the LLM is distilled into a lightweight routing model to ensure efficiency.
* Expert-aware Dynamic Pseudo-labeling: To solve the label scarcity problem, a flexible confidence threshold based on expert outputs is used to identify high-confidence pseudo-labeled nodes and utilize them for additional training.

To verify the effectiveness of the proposed framework, extensive experiments were conducted on multiple public benchmark datasets for domain generalization tasks, consistently demonstrating superior performance compared to existing methodologies.

**Strengths:**

* The problem is framed with a very valid and clear problem definition (Motivation) within the graph domain.
* The method was constructed to correspond with the Motivation. For example, to solve the heterogeneity and simplicity bias problems, the framework was designed so that MoE handles heterogeneity by utilizing a diverse expert pool, and an LLM-guided router, which is less sensitive to spurious patterns, handles simplicity bias.

**Weaknesses:**

* While the model's ultimate goal is to secure GNN generalizability, its high complexity and significant performance sensitivity to parameters suggest that adapting it to diverse datasets will likely incur substantial training costs.
* To validate the model's generalizability, experiments on more diverse domains beyond just citation networks seem necessary. Furthermore, it is necessary to evaluate its domain generalization performance on dissimilar domains, not just those with similar characteristics. However, the experiments are too limited to demonstrate this.

**Questions:**

* What are the advantages or disadvantages in training time and inference time compared to the baselines?
* How many iterations (B) are performed to select anchor nodes, and what is the performance sensitivity to this parameter?

---

### Note · Authors · 2026-01-06

I have read and agree with the venue's withdrawal policy on behalf of myself and my co-authors.